# Budgeted stream-based active learning
# via adaptive submodular maximization

**Kaito Fujii**
Kyoto University
JST, ERATO, Kawarabayashi Large Graph Project
fujii@ml.ist.i.kyoto-u.ac.jp

**Hisashi Kashima**
Kyoto University
kashima@i.kyoto-u.ac.jp

## Abstract

Active learning enables us to reduce the annotation cost by adaptively selecting unlabeled instances to be labeled. For pool-based active learning, several effective methods with theoretical guarantees have been developed through maximizing some utility function satisfying adaptive submodularity. In contrast, there have been few methods for stream-based active learning based on adaptive submodularity. In this paper, we propose a new class of utility functions, policy-adaptive submodular functions, which includes many existing adaptive submodular functions appearing in real world problems. We provide a general framework based on policy-adaptive submodularity that makes it possible to convert existing pool-based methods to stream-based methods and give theoretical guarantees on their performance. In addition we empirically demonstrate their effectiveness by comparing with existing heuristics on common benchmark datasets.

## 1 Introduction

Active learning is a problem setting for sequentially selecting unlabeled instances to be labeled, and it has been studied with much practical interest as an efficient way to reduce the annotation cost. One of the most popular settings of active learning is the pool-based one, in which the learner is given the entire set of unlabeled instances in advance, and iteratively selects an instance to be labeled next. The stream-based setting, which we deal with in this paper, is another important setting of active learning, in which the entire set of unlabeled instances are hidden initially, and presented one by one to the learner. This setting also has many real world applications, for example, sentiment analysis of web stream data [26], spam filtering [25], part-of-speech tagging [10], and video surveillance [23].

Adaptive submodularity [19] is an adaptive extension of submodularity, a natural diminishing return condition. It provides a framework for designing effective algorithms for several adaptive problems including pool-based active learning. For instance, the ones for noiseless active learning [19, 21] and the ones for noisy active learning [20, 9, 8] have been developed in recent years. Not only they have strong theoretical guarantees on their performance, but they perform well in practice compared with existing widely-used heuristics.

In spite of its considerable success in the pool-based setting, little is known about benefits of adaptive submodularity in the stream-based setting. This paper answers the question: is it possible to construct algorithms for stream-based active learning based on adaptive submodularity? We propose a general framework for creating stream-based algorithms from existing pool-based algorithms.

In this paper, we tackle the problem of stream-based active learning with a limited budget for making queries. The goal is collecting an informative set of labeled instances from a data stream of a certain length. The stream-based active learning problem has been typically studied in two settings:

the stream setting and the secretary setting, which correspond to memory constraints and timing constraints respectively; we treat both in this paper.

We formalize these problems as the adaptive stochastic maximization problem in the stream or secretary setting. For solving this problem, we propose a new class of stochastic utility functions: *policy-adaptive submodular functions*, which is another adaptive extension of submodularity. We prove this class includes many existing adaptive submodular functions used in various applications. Assuming the objective function satisfies policy-adaptive submodularity, we propose simple methods for each problem, and give theoretical guarantees on their performance in comparison to the optimal pool-based method. Experiments conducted on benchmark datasets show the effectiveness of our methods compared with several heuristics. Due to our framework, many algorithms developed in the pool-based setting can be converted to the stream-based setting.

In summary, our main contributions are the following:

- We provide a general framework that captures budgeted stream-based active learning and other applications.
- We propose a new class of stochastic utility functions, *policy-adaptive submodular functions*, which is a subclass of the adaptive submodular functions, and prove this class includes many existing adaptive submodular functions in real world problems.
- We propose two simple algorithms, `AdaptiveStream` and `AdaptiveSecretary`, and give theoretical performance guarantees on them.

## 2   Problem Settings

In this section, we first describe the general framework, then illustrate applications including stream-based active learning.

### 2.1   Adaptive Stochastic Maximization in the Stream and Secretary Settings

Here we specify the problem statement. This problem is a generalization of budgeted stream-based active learning and other applications.

Let $V = \{v_1, \cdots, v_n\}$ denote the entire set of $n$ items, and each item $v_i$ is in a particular state out of the set $\mathcal{Y}$ of possible states. Denote by $\phi : V \to \mathcal{Y}$ a realization of the states of the items. Let $\Phi$ be a random realization, and $Y_i$ a random variable representing the state of each item $v_i$ for $i = 1, \cdots, n$, i.e., $Y_i = \Phi(v_i)$. Assume that $\phi$ is generated from a known prior distribution $p(\phi)$. Suppose the state $Y_i$ is revealed when $v_i$ is selected. Let $\psi_A : A \to \mathcal{Y}$ denote the partial realization obtained after the states of items $A \subseteq V$ are observed. Note that a partial realization $\psi_A$ can be regarded as the set of observations $\{(s, \psi_A(s)) \mid s \in A\} \subseteq V \times \mathcal{Y}$.

We are given a set function[1] $f : 2^{V \times \mathcal{Y}} \to \mathbb{R}_{\geq 0}$ that defines the utility of observations made when some items are selected. Consider iteratively selecting an item to observe its state and aiming to make observations of high utility value. A policy $\pi$ is some decision tree that represents a strategy for adaptively selecting items. Formally it is defined to be a partial mapping that determines an item to be selected next from the observations made so far. Given some budget $k \in \mathbb{Z}_{>0}$, the goal is constructing a policy $\pi$ maximizing $\mathbb{E}_{\Phi}[f(\psi(\pi, \Phi))]$ subject to $|\psi(\pi, \phi)| \leq k$ for all $\phi$ where $\psi(\pi, \phi)$ denotes the observations obtained by executing policy $\pi$ under realization $\phi$.

This problem has been studied mainly in the pool-based setting, where we are given the entire set $V$ from the beginning and adaptively observe the states of items in any order. In this paper we tackle the stream-based setting, where the items are hidden initially and arrive one by one. The stream-based setting arises in two kinds of scenarios: one is the stream setting[2], in which we can postpone deciding whether or not to select an item by keeping it in a limited amount of memory, and at any time observe the state of the stored items. The other is the secretary setting, in which we must decide

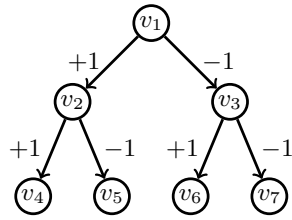

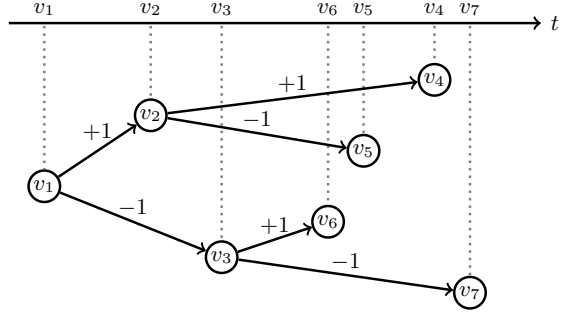

(a) A policy tree for the pool-based setting    (b) A policy tree for the stream-based setting

Figure 1: Examples of a pool-based policy and a stream-based policy in the case of $\mathcal{Y} = \{+1, -1\}$. (a) A pool-based policy can select items in an arbitrary order. (b) A stream-based policy must select items under memory or timing constraints taking account of only items that arrived so far.

immediately whether or not to select an item at each arrival. In both settings we assume the items arrive in a random order. The comparison of policies for the pool-based and stream-based settings is indicated in Figure 1.

## 2.2    Budgeted Stream-based Active Learning

We consider a problem setting called *Bayesian active learning*. Here $V$ represents the set of instances, $Y_1, \cdots, Y_n$ the initially unknown labels of the instances, and $\mathcal{Y}$ the set of possible labels.

Let $\mathcal{H}$ denote the set of candidates for the randomly generated true hypothesis $H$, and $p_H$ denote a prior probability over $\mathcal{H}$. When observations of the labels are noiseless, every hypothesis $h \in \mathcal{H}$ represents a particular realization, i.e., $h$ corresponds to some $\phi \in \mathcal{Y}^V$. When observations are noisy, the probability distribution $\mathbb{P}[Y_1, \cdots, Y_n | H = h]$ of the labels is not necessarily deterministic for each $h \in \mathcal{H}$. In both cases, we can iteratively select an instance and query its label to the annotation oracle. The objective is to determine the true hypothesis or one whose prediction error is small. Both the pool-based and stream-based settings have been extensively studied. The stream-based setting contains the stream and secretary settings, both of which have a lot of real world applications.

A common approach for devising a pool-based algorithm is designing some utility function that represents the informativeness of a set of labeled instances, and greedily selecting the instance maximizing this utility in terms of the expected value. We introduce the utility into stream-based active learning, and aim to collect $k$ labeled instances of high utility where $k \in \mathbb{Z}_{>0}$ is the budget on the number of queries. While most of the theoretical results for stream-based active learning are obtained assuming the data stream is infinite, we assume the length of the total data stream is given in advance.

## 2.3    Other Applications

We give a brief sketch of two examples that can be formalized as the adaptive stochastic maximization problem in the secretary setting. Both are variations for streaming data of the problems first proposed by Golovin and Krause [19].

One is adaptive viral marketing whose aim is spreading information about a new product through social networks. In this problem we adaptively select $k$ people to whom a free promotional sample of the product is offered so as to let them recommend the product to their friends. We cannot know if he recommends the product before actually offering a sample to each. The objective is maximizing the number of people that information of the product reaches. There arise some situations where people come sequentially, and at each arrival we must decide whether or not to offer a sample to them.

Another is adaptive sensor placement. We want to adaptively place $k$ unreliable sensors to cover the information obtained by them. The informativeness of each sensor is unknown before its deploy-

ment. We can consider the cases where the timing of placing sensors at each location is restricted for some reasons such as transportation cost.

## 3 Policy-Adaptive Submodularity

In this section, we discuss conditions satisfied by the utility functions of adaptive stochastic maximization problems.

Submodularity [17] is known as a natural diminishing return condition satisfied by various set functions appearing in a lot of applications, and adaptive submodularity was proposed by Golovin and Krause [19] as an adaptive extension of submodularity. Adaptive submodularity is defined as the diminishing return property about the expected marginal gain of a single item, i.e., $\Delta(s|\psi_A) \geq \Delta(s|\psi_B)$ for any partial realization $\psi_A \subseteq \psi_B$ and item $s \in V \setminus B$, where

$$\Delta(s|\psi) = \mathbb{E}_\Phi[f(\psi \cup \{(s, \Phi(s))\}) - f(\psi) \mid \Phi \sim p(\Phi|\psi)].$$

Similarly, adaptive monotonicity, an adaptive analog of monotonicity, is defined to be $\Delta(s|\psi_A) \geq 0$ for any partial realization $\psi_A$ and item $s \in V$. It is known that many utility functions used in the above applications satisfy the adaptive submodularity and the adaptive monotonicity. In the pool-based setting, greedily selecting the item of the maximal expected marginal gain yields $(1 - 1/e)$-approximation if the objective function is adaptive submodular and adaptive monotone [19].

Here we propose a new class of stochastic utility functions, *policy-adaptive submodular functions*. Let $\mathrm{range}(\pi)$ denote the set containing all items that $\pi$ selects for some $\phi$, and we define policy-adaptive submodularity as the diminishing return property about the expected marginal gain of any policy as follows.

**Definition 3.1** (Policy-adaptive submodularity). *A set function $f : 2^{V \times \mathcal{Y}} \to \mathbb{R}_{\geq 0}$ is policy-adaptive submodular with respect to a prior distribution $p(\phi)$, or $(f, p)$ is policy-adaptive submodular, if $\Delta(\pi|\psi_A) \geq \Delta(\pi|\psi_B)$ holds for any partial realization $\psi_A$, $\psi_B$ and policy $\pi$ such that $\psi_A \subseteq \psi_B$ and $\mathrm{range}(\pi) \subseteq V \setminus B$, where*

$$\Delta(\pi|\psi) = \mathbb{E}_\Phi[f(\psi \cup \psi(\pi, \Phi)) - f(\psi) \mid \Phi \sim p(\Phi|\psi)].$$

Since a single item can be regarded as a policy selecting only one item, policy-adaptive submodularity is a stricter condition than adaptive submodularity.

Policy-adaptive submodularity is also a natural extension of submodularity. The submodularity of a set function $f : 2^V \to \mathbb{R}_{\geq 0}$ is defined as the condition that $f(A \cup \{s\}) - f(A) \geq f(B \cup \{s\}) - f(B)$ for any $A \subseteq B \subseteq V$ and $s \in V \setminus B$, which is equivalent to the condition that $f(A \cup P) - f(A) \geq f(B \cup P) - f(B)$ for any $A \subseteq B \subseteq V$ and $P \subseteq V \setminus B$. Adaptive extensions of these conditions are adaptive submodularity and policy-adaptive submodularity respectively. Nevertheless there is a counterexample to the equivalence of adaptive submodularity and policy-adaptive submodularity, which is given in the supplementary materials.

Surprisingly, many existing adaptive submodular functions in applications also satisfy the policy-adaptive submodularity. In active learning, the objective function of generalized binary search [12, 19], EC$^2$ [20], ALuMA [21], and the maximum Gibbs error criterion [9, 8] are not only adaptive submodular, but policy-adaptive submodular. In other applications including influence maximization and sensor placements, it is often assumed that the variables $Y_1, \cdots, Y_n$ are independent, and the policy-adaptive submodularity always holds in this case. The proofs of these propositions are given in the supplementary materials.

To give the theoretical guarantees for the algorithms introduced in the next section, we assume not only the adaptive submodularity and the adaptive monotonicity, but also the policy-adaptive submodularity. However, our theoretical analyses can still be applied to many applications.

## 4 Algorithms

In this section we describe our proposed algorithms for each of the stream and secretary settings, and state the theoretical guarantees on their performance. The full versions of pseudocodes are given in the supplementary materials.

---
**Algorithm 1** `AdaptiveStream` algorithm & `AdaptiveSecretary` algorithm
---
**Input:** A set function $f : 2^{V \times \mathcal{Y}} \to \mathbb{R}_{\geq 0}$ and a prior distribution $p(\phi)$ such that $(f, p)$ is policy-adaptive submodular and adaptive monotone. The number of items in the entire stream $n \in \mathbb{Z}_{>0}$. A budget $k \in \mathbb{Z}_{>0}$. Randomly permuted stream of the items, denoted by $(s_1, \cdots, s_n)$.
**Output:** Some observations $\psi_k \subseteq V \times \mathcal{Y}$ such that $|\psi_k| \leq k$.
  1: Let $\psi_0 := \emptyset$.
  2: **for** each segment $S_l = \{s_i \mid (l-1)n/k < i \leq ln/k\}$ **do**
  3:      Select an item $s$ out of $S_l$ by
       $\begin{cases} \text{selecting the item of the largest expected marginal gain} & (\texttt{AdaptiveStream}) \\ \text{applying the classical secretary algorithm} & (\texttt{AdaptiveSecretary}) \end{cases}$
  4:      Observe the state $y$ of item $s$ and let $\psi_l := \psi_{l-1} \cup \{(s, y)\}$.
  5: **return** $\psi_k$ as the solution
---

## 4.1 Algorithm for the Stream Setting

The main idea of our proposed method is simple: divide the entire stream into $k$ segments and select the best item from each one. For simplicity, we consider the case where $n$ is a multiple integer of $k$. If $n$ is not, we can add $k\lceil \frac{n}{k} \rceil - n$ dummy items with no benefit and prove the same guarantee. Our algorithm first divides the item sequence $s_1, \cdots, s_n$ into $S_l = \{s_i \mid (l-1)n/k < i \leq ln/k\}$ for $l = 1, \cdots, k$. In each segment, the algorithm selects the item of the largest expected marginal gain, that is, $\operatorname{argmax}\{\Delta(s|\psi_{l-1}) \mid s \in S_l\}$ where $\psi_{l-1}$ is the partial realization obtained before the $l$th segment. This can be implemented with only $\mathrm{O}(1)$ space by storing only the item of the maximal expected marginal gain so far in the current segment. We provide the theoretical guarantee on the performance of this algorithm by utilizing the policy-adaptive submodularity of the objective function.

**Theorem 4.1.** *Suppose $f : 2^{V \times \mathcal{Y}} \to \mathbb{R}_{\geq 0}$ is policy-adaptive submodular and adaptive monotone w.r.t. a prior $p(\phi)$. Assume the items come sequentially in a random order. For any policy $\pi$ such that $|\psi(\pi, \phi)| \leq k$ holds for all $\phi$,* `AdaptiveStream` *selects $k$ items using $\mathrm{O}(1)$ space and achieves at least $0.16$ times the expected total gain of $\pi$ in expectation.*

## 4.2 Algorithm for the Secretary Setting

Though our proposed algorithm for the secretary setting is similar in its approach to the one for the stream setting, it is impossible to select the item of the maximal expected marginal gain from each segment in the secretary setting. Then we use *classical secretary algorithm* [13] as a subroutine to obtain the maximal item at least with some constant probability. The classical secretary algorithm lets the first $\lfloor n/(ek) \rfloor$ items pass and then selects the first item whose value is larger than all items so far. The probability that this subroutine selects the item of the largest expected marginal gain is at least $1/e$ at each segment. This algorithm can be viewed as an adaptive version of the algorithm for the monotone submodular secretary problem [3]. We give the guarantee similar to the one for the stream setting.

**Theorem 4.2.** *Suppose $f : 2^{V \times \mathcal{Y}} \to \mathbb{R}_{\geq 0}$ is policy-adaptive submodular and adaptive monotone w.r.t. a prior $p(\phi)$. Assume the items come sequentially in a random order. For any policy $\pi$ such that $|\psi(\pi, \phi)| \leq k$ holds for all $\phi$,* `AdaptiveSecretary` *selects at most $k$ items and achieves at least $0.08$ times the expected total gain of $\pi$ in expectation.*

# 5 Overview of Theoretical Analysis

In this section we briefly describe the proofs of Theorem 4.1 and 4.2, and compare our techniques with the previous work. The full proofs are given in the supplementary materials.

The methods used in the proofs of both theorems are almost the same. They consist of two steps: in the first step, we bound the expected marginal gain of each item and in the second step, we take summation of one step marginal gains and derive the overall bound for the algorithms. Though our techniques used in the second step are taken from the previous work [3], the first step contains several novel techniques.

Let $\Delta_i$ be the expected marginal gain of an item picked from the $i$th segment $S_i$. First we bound it from below with the difference between the optimal pool-based policy $\pi_T^*$ for selecting $k$ items from $T$ and the policy $\pi_{i-1}^\sigma$ that encodes the algorithm until $i-1$th step under a permutation $\sigma$ in which the items arrive. For the non-adaptive setting, the items in the optimal set are distributed among the segments uniformly at random, then we can evaluate $\Delta_i$ by considering whether $S_i$ contains an item included in the optimal set [3]. On the other hand, in the adaptive setting, it is difficult to consider how $\pi_T^*$ is distributed in the unarrived items because the policy is closely related not only to the contained items but also to the order of items. Then we compare $\Delta_i$ and the marginal gain of $\pi_T^*$ directly. With the adaptive monotonicity, we obtain $\Delta_i \geq (1 - \exp(-\frac{k}{k-i+1}))(f_{\mathrm{avg}}(\pi_T^*) - f_{\mathrm{avg}}(\pi_{i-1}^\sigma))/k$ where $f_{\mathrm{avg}}(\pi) = \mathbb{E}_\Phi[f(\psi(\pi, \Phi))]$.

Next we bound $f_{\mathrm{avg}}(\pi_T^*)$ with the optimal pool-based policy $\pi_V^*$ that selects $k$ items from $V$. For the non-adaptive setting, we can apply a widely-used lemma proved by Feige, Mirrokni, and Vondrák [15]. This lemma provides a bound for the expected value of a randomly deleted subset. To extend this lemma to the adaptive setting, we define a partially deleted policy tree, *grafted policy*, and prove the adaptive version of the lemma with the policy-adaptive submodularity. From this lemma we can obtain the bound $\mathbb{E}_\sigma[f_{\mathrm{avg}}(\pi_T^*)] \geq (k-i+1)f_{\mathrm{avg}}(\pi_V^*)/k$. We also provide an example that shows adaptive submodularity is not enough to prove this lemma.

Summing the bounds for each one-step expected marginal gain until $l$th step ($l$ is specified in the full proof for optimizing the resulting guarantees), we can conclude that our proposed algorithms achieve some constant factor approximation in comparison to the optimal pool-based policy. Though `AdaptiveSecretary` is the adaptive version of the existing algorithm, our resulting constant factor is a little worse than the original $(1-1/\mathrm{e})/7$ due to the above new analyses.

# 6 Experiments

## 6.1 Experimental Setting

We conducted experiments on budgeted active learning in the following three settings: the pool-based, stream, and secretary settings. For each setting, we compare two methods: one is based on the policy-adaptive submodularity and the other is based on *uncertainty sampling* as baselines. Uncertainty sampling is the most widely-used approach in applications. Selecting random instances, which we call random, is also implemented as another baseline that can be used in every setting.

We select ALuMA [21] out of several pool-based methods based on adaptive submodularity, and convert it to the stream and secretary settings with `AdaptiveStream` and `AdaptiveSecretary`, which we call stream submodular and secretary submodular respectively. For comparison, we also implement the original pool-based method, which we call pool submodular. Though ALuMA is designed for the noiseless case, there is a modification method that makes its hypotheses sampling more noise-tolerant [7], which we employ. The number of hypotheses sampled at each time is set $N = 1000$ in all settings.

For the pool-based setting, uncertainty sampling is widely-known as a generic and easy-to-implement heuristic in many applications. This selects the most uncertain instance, i.e., the instance that is closest to the current linear separator. In contrast, there is no standard heuristic for the stream and secretary settings. We apply the same conversion to the pool-based uncertain sampling method as `AdaptiveStream` and `AdaptiveSecretary`, i.e., in the stream setting, selecting the most uncertain instance from the segment at each step, and in the secretary setting, running the classical secretary algorithm to select the most uncertain instance at least with probability $1/\mathrm{e}$. A similar one to this approach in the stream setting is used in some applications [26]. In every setting, we first randomly select 10 instances for the initial training of a classifier and after that, select $k-10$ instances with each method. We use the linear SVM trained with instances labeled so far to judge the uncertainty. We call these methods pool uncertainty, stream uncertainty, secretary uncertainty respectively, and use them as baselines.

We conducted experiments on two benchmark datasets, WDBC[3] and MNIST[4]. The WDBC dataset contains 569 instances, each of which consists of 32-dimensional features of cells and their diagnosis

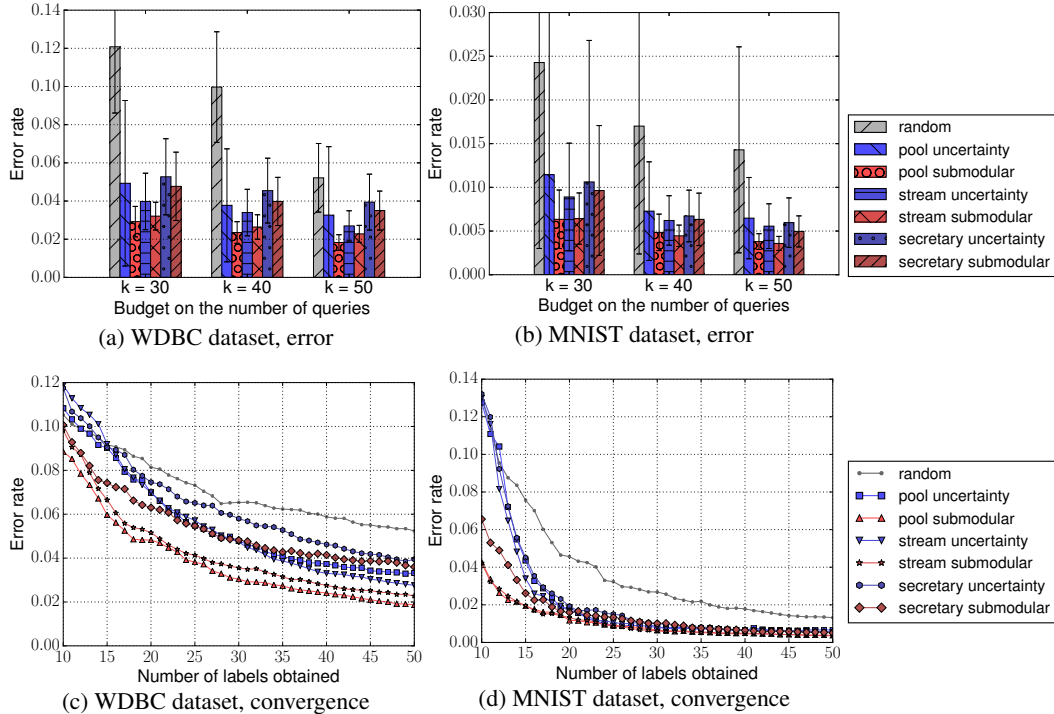

Figure 2: Experimental results

results. From the MNIST dataset, the dataset of handwritten digits, we extract 14780 images of the two classes, 0 and 1, so as to consider the binary classification problem, and apply PCA to reduce its dimensions from 784 to 10. We standardize both datasets so that the values of each feature have zero mean and unit variance.

We evaluate the performance through 100 trials, where at each time an order in which the instances arrive is generated randomly. For all the methods, we calculate the error rate by training linear SVM with the obtained labeled instances and testing with the entire dataset.

### 6.2 Experimental Results

Figure 2(a)(b) illustrate the average error rate achieved by each method with budget $k = 30, 40, 50$. Our methods stream submodular and secretary submodular outperform not only random, but also stream uncertainty and secretary uncertainty respectively, i.e., the methods based on policy-adaptive submodularity perform better than the methods based on uncertainty sampling in each of the stream and secretary settings. Moreover, we can observe our methods are stabler than the other methods from the error bars representing the standard deviation.

Figure 2(c)(d) show how the error rate decreases as labels are queried in the case of $k = 50$. In both datasets, we can observe the performance of stream submodular is competitive with pool submodular.

## 7 Related Work

**Stream-based active learning.** Much amount of work has been dedicated to devising algorithms for stream-based active learning (also known as *selective sampling*) from both the theoretical and practical aspects. From the theoretical aspects, several bounds on the label complexity have been provided [16, 2, 4], but their interest lies in the guarantees compared to the passive learning, not the optimal algorithm. From the practical aspects, it has been applied to many real world problems such as sentiment analysis of web stream data [26], spam filtering [25], part-of-speech tagging [10], and video surveillance [23], but there is no definitive widely-used heuristic.

Of particular relevance to our work is the one presented by Sabato and Hess [24]. They devised general methods for constructing stream-based algorithms satisfying a budget based on pool-based algorithms, but their theoretical guarantees are bounding the length of the stream needed to emulate the pool-based algorithm, which is a large difference from our work. Das et al. [11] designed the algorithm for adaptively collecting water samples, referring to the submodular secretary problem, but they focused on applications to marine ecosystem monitoring, and did not give any theoretical analysis about its performance.

**Adaptive submodular maximization.** The framework of adaptive submodularity, which is an adaptive counterpart of submodularity, is established by Golovin and Krause [19]. It provides the simple greedy algorithm with the near-optimal guarantees in several adaptive real world problems. Specifically it achieves remarkable success in pool-based active learning. For the noiseless cases, Golovin and Krause [19] described the generalized binary search algorithm [12] as the greedy algorithm for some adaptive submodular function, and improved its approximation factor. Golovin et al. [20] provided an algorithm for Bayesian active learning with noisy observations by reducing it to the equivalence class determination problem. On the other hand, there have been several studies on adaptive submodular maximization in other settings, for example, selecting multiple instances at the same time before observing their states [7], guessing an unknown prior distribution in the bandit setting [18], and maximizing non-monotone adaptive submodular functions [22].

**Submodular maximization in the stream and secretary settings.** Submodular maximization in the stream setting, called *streaming submodular maximization*, has been studied under several constraints. Badanidiyuru et al. [1] provided a $(1/2 - \epsilon)$-approximation algorithm that can be executed in $\mathrm{O}(k \log k)$ space for the cardinality constraint. For more general constraints including matching and multiple matroids constraints, Chakrabarti and Kale [5] proposed constant factor approximation algorithms. Chekuri et al. [6] devised algorithms for non-monotone submodular functions.

On the other hand, much effort is also devoted to submodular maximization in the secretary setting, called *submodular secretary problem*, under various constraints. Bateni et al. [3] specified the problem first and provided algorithms for both monotone and non-monotone submodular secretary problems under several constraints, one of which our methods are based on. Feldman et al. [14] improved constant factors of the theoretical guarantees for monotone cases.

# 8   Concluding Remarks

In this paper, we investigated stream-based active learning with a budget constraint in the view of adaptive submodular maximization. To tackle this problem, we introduced the adaptive stochastic maximization problem in the stream and secretary settings, which can formalize stream-based active learning. We provided a new class of objective functions, policy-adaptive submodular functions, and showed this class contains many utility functions that have been used in pool-based active learning and other applications. `AdaptiveStream` and `AdaptiveSecretary`, which we proposed in this paper, are simple algorithms guaranteed to be constant factor competitive with the optimal pool-based policy. We empirically demonstrated their performance by applying our algorithms to the budgeted stream-based active learning problem, and our experimental results indicate their effectiveness compared to the existing methods.

There are two natural directions for future work. One is exploring the possibility of the concept, policy-adaptive submodularity. By studying the nature of this class, we can probably yield theoretical insight for other problems. Another is further developing the practical aspects of our results. In real world problems sometimes it happens that the items arrive not in a random order. For example, in sequential adaptive sensor placement [11], an order of items is restricted to some transportation constraint. In this setting our guarantees do not hold and another algorithm is needed. In contrast to the non-adaptive setting, even in the stream setting, it seems much more difficult to design a constant factor approximation algorithm because the full information of each item is totally revealed when its state is observed and memory is not so powerful as in the non-adaptive setting.

**Acknowledgments**

The second author is supported by Grant-in-Aid for Scientific Research on Innovative Areas, Exploration of nanostructure-property relationships for materials innovation.

## Footnotes

[1] In the original definition of stochastic utility functions [19], the objective value depends not only on the partial realization $\psi$, but also on the realization $\phi$. However, given such $f : 2^V \times \mathcal{Y}^V \to \mathbb{R}_{\geq 0}$, we can redefine $\tilde{f} : 2^{V \times \mathcal{Y}} \to \mathbb{R}_{\geq 0}$ as $\tilde{f}(\psi_A) = \mathbb{E}_{\Phi}[f(A, \Phi) \mid \Phi \sim p(\Phi|\psi_A)]$, and it does not critically change the overall discussion in our problem settings. Thus for notational convenience, we use the simpler definition.

[2] In this paper, "stream-based setting" and "stream setting" are distinguished.

[3] https://archive.ics.uci.edu/ml/datasets/Breast+Cancer+Wisconsin+(Diagnostic)

[4] http://yann.lecun.com/exdb/mnist/

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
