[Supplementary Material · nips_2016.pdf]

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

[5]The strong adaptive monotonicity of $(f, p)$ is defined to be $\psi \subseteq \psi' \Rightarrow f(\psi) \leq f(\psi')$.

[6]Here we must extend the definition of the policy to randomized policies. A randomized policy is a partial mapping from the current partial realization to a distribution over items that represents the item to be selected next. It is easy to see that the optimal deterministic policy is still optimal among randomized policies under any budget since a randomized policy can be expressed as the weighted sum of deterministic policies.

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

# A  Proofs of the Policy-Adaptive Submodularity

In this section, we show several adaptive submodular functions used as utility functions in applications are policy-adaptive submodular.

Let $\mathcal{H}$ denote the set of candidates for the true hypothesis. First we consider the noiseless case. Each hypothesis $h \in \mathcal{H}$ represents some realization, i.e., $h : V \to \mathcal{Y}$. Let $p_H$ be a prior distribution over hypotheses. For convenience, we use $p_H$ as the sum of the probability $p_H(\mathcal{H}') = \sum_{h \in \mathcal{H}'} p_H(h)$ for any $\mathcal{H}' \subseteq \mathcal{H}$. We can obtain the prior distribution over realizations as $p(\phi) = p_H(\{h \mid \forall s \in V, h(s) = \phi(s)\})$. The version space under observations $\psi$ is defined to be $\mathcal{H}(\psi) = \{h \in \mathcal{H} \mid \forall (s, y) \in \psi, h(s) = y\}$.

**The objective function of generalized binary search.** Generalized binary search in the Bayesian setting [12], designed for noiseless active learning, is executed by greedily maximizing

$$f_{\mathrm{GBS}}(\psi) = 1 - p_H(\mathcal{H}(\psi)).$$

Golovin and Krause [19] proved this function is adaptive submodular, and improved the upper bound on the number of labels made by generalized binary search for deciding the true hypothesis.

**Proposition A.1.** $f_{\mathrm{GBS}}$ *is policy-adaptive submodular w.r.t.* $p(\phi)$.

*Proof.* Let $\psi_0$ be any partial realization and $\pi$ any policy such that $\mathrm{range}(\pi) \subseteq V \setminus \mathrm{dom}(\psi_0)$. The set of all resulting observations of $\pi$ is defined to be $\mathcal{L}(\pi) = \{\psi \mid \exists \phi, \psi(\pi, \phi)\}$. The expected marginal gain of $\pi$ under the observations $\psi_0$ is

$$\Delta(\pi|\psi_0) = \sum_{\psi \in \mathcal{L}(\pi)} p(\psi|\psi_0)\Big(f_{\mathrm{GBS}}(\psi \cup \psi_0) - f_{\mathrm{GBS}}(\psi_0)\Big)$$

$$= \sum_{\psi \in \mathcal{L}(\pi)} \frac{p_H(\mathcal{H}(\psi \cup \psi_0))}{p_H(\mathcal{H}(\psi_0))}\Big(p_H(\mathcal{H}(\psi_0)) - p_H(\mathcal{H}(\psi \cup \psi_0))\Big).$$

Let $g$ be a function of $\mathbf{x} = (x_\psi)_\psi = (p_H(\mathcal{H}(\psi \cup \psi_0)))_\psi$ such that $g(\mathbf{x}) = \Delta(\pi|\psi_0)$, then $g(\mathbf{x}) = \sum_\psi x_\psi - \sum_\psi x_\psi^2 / \sum_\psi x_\psi$. We show this function is monotonically increasing about each $x_\psi$.

By differentiating $g$ with respect to $x_\psi$,

$$\frac{\partial}{\partial x_\psi} g(\mathbf{x}) = 1 + \frac{\sum_{\psi'} x_{\psi'}^2}{(\sum_{\psi'} x_{\psi'})^2} - \frac{2x_\psi}{\sum_{\psi'} x_{\psi'}} = \left(1 - \frac{x_\psi}{\sum_{\psi'} x_{\psi'}}\right)^2 + \frac{\sum_{\psi' \neq \psi} x_{\psi'}^2}{(\sum_{\psi'} x_{\psi'})^2}$$

which is at least 0. Therefore $g$ is monotonically increasing w.r.t. each $x_\psi$.

If $\psi_A$ and $\psi_B$ are partial realizations such that $\psi_A \subseteq \psi_B$ and $\mathrm{range}(\pi) \subseteq V \setminus B$, it holds that $p_H(\mathcal{H}(\psi \cup \psi_A)) \geq p_H(\mathcal{H}(\psi \cup \psi_B))$ for all $\psi \in \mathcal{L}(\pi)$, which implies $\Delta(\pi|\psi_A) \geq \Delta(\pi|\psi_B)$.  $\square$

**The objective function of EC$^2$ algorithm.** Suppose that $\mathcal{H}$ is partitioned into equivalence classes $\mathcal{H}_1, \cdots, \mathcal{H}_m$, i.e., $\mathcal{H}_1 \cup \cdots \cup \mathcal{H}_m = \mathcal{H}$ and $\mathcal{H}_i \cap \mathcal{H}_j = \emptyset$ for every $i \neq j$. The equivalence class determination problem [20] is the problem of deciding which equivalence class the true hypothesis lies in. Golovin et al. [20] proves that

$$f_{\mathrm{EC}}(\psi) = 1 - \sum_{i<j}\Big(p_H(\mathcal{H}_i(\psi))\, p_H(\mathcal{H}_j(\psi))\Big)$$

is adaptive submodular and adaptive monotone, and this function achieves the value 1 if and only if the equivalence class that the true hypothesis lies in is determined. Now we show $(f_{\mathrm{EC}}, p)$ is policy-adaptive submodular.

**Proposition A.2.** $f_{\mathrm{EC}}$ *is policy-adaptive submodular w.r.t.* $p(\phi)$.

*Proof.* Let $\psi_0$ be any partial realization and $\pi$ any policy such that $\mathrm{range}(\pi) \subseteq V \setminus \mathrm{dom}(\psi_0)$. The set of all resulting observations of $\pi$ is defined to be $\mathcal{L}(\pi) = \{\psi \mid \exists \phi, \psi(\pi, \phi)\}$. The expected

marginal gain of $\pi$ under the partial realization $\psi_0$ is

$$\Delta(\pi|\psi_0) = \sum_{\psi \in \mathcal{L}(\pi)} p(\psi|\psi_0)\Big(f_{\mathrm{EC}}(\psi \cup \psi_0) - f_{\mathrm{EC}}(\psi_0)\Big)$$

$$= \sum_{\psi \in \mathcal{L}(\pi)} \frac{p_H(\mathcal{H}(\psi \cup \psi_0))}{p_H(\mathcal{H}(\psi_0))} \left( \sum_{i<j} \Big( p_H(\mathcal{H}_i(\psi_0))p_H(\mathcal{H}_j(\psi_0)) \Big) \right.$$

$$\left. - \sum_{i<j} \Big( p_H(\mathcal{H}_i(\psi \cup \psi_0))p_H(\mathcal{H}_j(\psi \cup \psi_0)) \Big) \right).$$

Let $g$ be a function of $\mathbf{x} = (x_{i,\psi})_{i,\psi} = (p_H(\mathcal{H}_i(\psi \cup \psi_0)))_{i,\psi}$ such that $g(\mathbf{x}) = \Delta(\pi|\psi_0)$, then

$$g(\mathbf{x}) = \sum_{\psi} \left[ \frac{\sum_i x_{i,\psi}}{\sum_{i,\psi'} x_{i,\psi'}} \sum_{i<j} \left( \left( \sum_{\psi'} x_{i,\psi'} \right) \left( \sum_{\psi'} x_{j,\psi'} \right) - x_{i,\psi}x_{j,\psi} \right) \right].$$

In the same way as the proof of the adaptive submodularity of $f_{\mathrm{EC}}$ given by Golovin et al. [20], we can see this function is monotonically increasing about each $x_{i,\psi}$ by showing the non-negativity of the derivative of $g$.

If $\psi_A$ and $\psi_B$ are partial realizations such that $\psi_A \subseteq \psi_B$ and $\mathrm{range}(\pi) \subseteq V \setminus B$, it holds that $p_H(\mathcal{H}_i(\psi \cup \psi_A)) \geq p_H(\mathcal{H}_i(\psi \cup \psi_B))$ for all $i = 1, \cdots, m$ and $\psi \in \mathcal{L}(\pi)$, which implies $\Delta(\pi|\psi_A) \geq \Delta(\pi|\psi_B)$. $\qquad\square$

**The objective function of ALuMA algorithm.** We consider the binary classification problem, i.e., $\mathcal{Y} = \{+1, -1\}$, for linearly separable instances. Suppose the parameter $\mathbf{w} \in \mathbb{R}^d$ is uniformly distributed in the $d$-dimensional unit ball $\mathbb{B}_1^d$, and the classifier of parameter $\mathbf{w}$ classifies an instance $\mathbf{x} \in \mathbb{R}^d$ into the class $\mathrm{sgn}(\langle \mathbf{w}, \mathbf{x} \rangle)$. Gonen et al. [21] proposed the objective function

$$f_{\mathrm{ALuMA}}(\psi) = 1 - \mathbb{P}[\{\mathbf{w} \in \mathbb{B}_1^d \mid \forall(\mathbf{x}, y) \in \psi, y = \mathrm{sgn}(\langle \mathbf{w}, \mathbf{x} \rangle)\}],$$

and prove it is adaptive submodular. The prior probability over realizations is $p(\phi) = \mathbb{P}[\{\mathbf{w} \in \mathbb{B}_1^d \mid \forall \mathbf{x} \in V, \Phi(\mathbf{x}) = \mathrm{sgn}(\langle \mathbf{w}, \mathbf{x} \rangle)\}]$.

**Proposition A.3.** *The objective function $f_{\mathrm{ALuMA}}$ is policy-adaptive submodular w.r.t. $p(\phi)$.*

*Proof.* It is a special case of $f_{\mathrm{GBS}}$ where the prior probability $p(\phi)$ can be expressed as above. Hence $f_{\mathrm{ALuMA}}$ is policy-adaptive submodular. $\qquad\square$

Though running exactly the greedy algorithm for $f_{\mathrm{ALuMA}}$ is difficult, there is a method to approximately execute the greedy policy within some constant factor [21]. We can apply this approximate method to the stream-based setting and give the performance guarantees by using the policy-adaptive submodularity.

**The objective function of maximum Gibbs error criterion.** Now suppose labels are generated from noisy distribution, i.e., $\mathbb{P}[Y_1, \cdots, Y_n | H = h]$ is not deterministic. In this case, the prior over realizations is expressed as $p(\phi) = \sum_{h \in \mathcal{H}} p_H(h)\mathbb{P}[\phi|H = h]$. By using this distribution, the utility function of the maximum Gibbs error criterion [9, 8] is defined as

$$f_{\mathrm{Gibbs}}(\psi) = 1 - p(\psi).$$

**Proposition A.4.** $f_{\mathrm{Gibbs}}$ *is policy-adaptive submodular w.r.t. $p(\phi)$.*

*Proof.* The same proof as the one of Proposition A.1 can be applied by replacing the probability mass of hypotheses $p_H(\mathcal{H}(\psi))$ with the probability mass of partial realizations $p(\psi)$. $\qquad\square$

**In cases where the states of the items are independent.** In many applications other than active learning, it is often assumed that the states of the items are independent of each other. For instance, this assumption is often used in adaptive variations of sensor placements and influence maximization, both of which are introduced by Golovin and Krause [19]. In this case, the policy-adaptive submodularity follows from the adaptive submodularity.

**Proposition A.5.** *If $Y_1, \cdots, Y_n$ are independent and $(f, p)$ is adaptive submodular, then $(f, p)$ is policy-adaptive submodular.*

*Proof.* Let $\psi_A$ and $\psi_B$ be any partial realization such that $\psi_A \subseteq \psi_B$, and let $\pi$ denote any policy such that $\mathrm{range}(\pi) \subseteq V \setminus B$. We prove $\Delta(\pi|\psi_A) \geq \Delta(\pi|\psi_B)$ with the induction on the height of policy tree $\pi$.

In the case where the height of $\pi$ is 1, i.e., $\pi$ is a policy selecting only a single item, the statement follows from the adaptive submodularity of $(f, p)$.

Assume that the statement holds for all policies whose height is at most $l - 1$. Suppose the height of $\pi$ is $l$. Let $s$ denote the first item that $\pi$ selects and $\pi_y$ be the subpolicy of $\pi$ that is executed just after $Y_s$ turns out to be $y$ for each $y \in \mathcal{Y}$. Since $Y_1, \cdots, Y_n$ are independent and $s \in V \setminus B$, it holds that $p(Y_s = y|\psi_A) = p(Y_s = y|\psi_B)$. Applying the induction hypothesis to $\pi_y$, whose height is at most $l - 1$, we obtain $\Delta(\pi_y|\psi_A \cup \{(s, y)\}) \geq \Delta(\pi_y|\psi_B \cup \{(s, y)\})$. Therefore, we can see that

$$\Delta(\pi|\psi_A) = \Delta(s|\psi_A) + \sum_{y \in \mathcal{Y}} p(Y_s = y|\psi_A)\Delta(\pi_y|\psi_A \cup \{(s, y)\})$$

$$\geq \Delta(s|\psi_B) + \sum_{y \in \mathcal{Y}} p(Y_s = y|\psi_B)\Delta(\pi_y|\psi_B \cup \{(s, y)\})$$

$$= \Delta(\pi|\psi_B)$$

$\square$

**A counterexample to the equivalence.** As described above, many adaptive submodular objective functions proposed in previous work are also policy-adaptive submodular. However, there is a simple counterexample to the equivalence of adaptive submodularity and policy-adaptive submodularity.

**Example A.6.** Let $V = \{a, b, c\}$ and $\mathcal{Y} = \{+1, -1\}$. If $w : V \to \mathbb{R}_{\geq 0}$ is a weight vector such that $w(a) = w(b) = w(c) = 1$, the sum of weights $f(\psi) = \sum_{s \in \mathrm{dom}(\psi)} w(s)$ is adaptive submodular w.r.t. any prior distribution $p(\phi)$.

Let $\pi$ denote a policy that selects $b$ first, and proceeds to select $c$ if $Y_b = +1$. Assume the marginal distribution of $\{Y_a, Y_b\}$ is defined as $\mathbb{P}[Y_a = +1, Y_b = +1] = 0.09$, $\mathbb{P}[Y_a = +1, Y_b = -1] = \mathbb{P}[Y_a = -1, Y_b = +1] = 0.01$, and $\mathbb{P}[Y_a = -1, Y_b = -1] = 0.89$. Since $\Delta(\pi|\{(a, +1)\}) = 1.9$ is larger than $\Delta(\pi|\emptyset) = 1.1$, we know that $(f, p)$ is not policy-adaptive submodular.

$(f, p)$ is not only adaptive submodular, but linear and strongly adaptive monotone[5], hence we can see that policy-adaptive submodularity does not follow from any combination of these conditions.

## B  Pseudocodes of the Proposed Algorithms

---
**Algorithm 2** `AdaptiveStream` algorithm
---
**Input:** A set function $f : 2^{V \times \mathcal{Y}} \to \mathbb{R}_{\geq 0}$ and a prior distribution $p(\phi)$ such that $(f, p)$ is policy-adaptive submodular and adaptive monotone. The number of items in the entire stream $n \in \mathbb{Z}_{>0}$. A budget $k \in \mathbb{Z}_{>0}$. Randomly permuted stream of the items, denoted by $(s_1, \cdots, s_n)$.
**Output:** Some observations $\psi_k \subseteq V \times \mathcal{Y}$ such that $|\psi_k| \leq k$.
1: Let $\psi_0 := \emptyset$.
2: **for** each segment $S_l = \{s_i \mid (l - 1)n/k < i \leq ln/k\}$ **do**
3:     $\delta_{\max} := 0$
4:     **for** each item $s \in S_l$ **do**
5:         **if** $\Delta(s|\psi_{l-1}) \geq \delta_{\max}$ **then**
6:             $s_{\max} := s, \delta_{\max} := \Delta(s|\psi_{l-1})$.
7:     Observe the state $y$ of item $s_{\max}$ and let $\psi_l := \psi_{l-1} \cup \{(s_{\max}, y)\}$.
8: **return** $\psi_k$ as the solution
---

**Algorithm 3** `AdaptiveSecretary` algorithm

---

**Input:** A set function $f : 2^{V \times \mathcal{Y}} \to \mathbb{R}_{\geq 0}$ and a prior distribution $p(\phi)$ such that $(f, p)$ is policy-adaptive submodular and adaptive monotone. The number of items in the entire stream $n \in \mathbb{Z}_{>0}$. A budget $k \in \mathbb{Z}_{>0}$. Randomly permuted stream of the items, denoted by $(s_1, \cdots, s_n)$.
**Output:** Some observations $\psi_k \subseteq V \times \mathcal{Y}$ such that $|\psi_k| \leq k$.

1: Let $\psi_0 := \emptyset$.
2: **for** each segment $S_l = \{s_i \mid (l-1)n/k < i \leq ln/k\}$ **do**
3:      Let the first $\lfloor n/(ek) \rfloor$ items in $S_l$ pass,
     and let $\delta_{\max} := \{\Delta(s_i|\psi_{l-1}) \mid (l-1)n/k < i \leq (l-1)n/k + \lfloor n/(ek) \rfloor \}$.
4:      select := **false**
5:      **for** each $s_i$ such that $(l-1)n/k + \lfloor n/(ek) \rfloor < i \leq ln/k$ **do**
6:          **if** $\Delta(s_i|\psi_{l-1}) \geq \delta_{\max}$ **then**
7:              Observe the state $y$ of item $s_i$ and let $\psi_l := \psi_{l-1} \cup \{(s_i, y)\}$.
8:              select := **true**
9:              **break**
10:      **if** select = **false then**
11:          $\psi_l := \psi_{l-1}$.
12: **return** $\psi_k$ as the solution

---

## C  Proofs of the Theoretical Guarantees of the Algorithms

Define the conditional function as $f(\psi'|\psi) = f(\psi' \cup \psi) - f(\psi)$ where $\psi$ and $\psi'$ are partial realizations. We denote the expected marginal gain about function $f$ by $\Delta_f$. We can show the policy-adaptive submodularity is preserved by restriction as follows.

**Lemma C.1.** *Suppose $f : 2^{V \times \mathcal{Y}} \to \mathbb{R}$ is policy-adaptive submodular w.r.t. a prior $p(\phi)$. For an arbitrary partial realization $\psi_S$, $f(\cdot|\psi_S)$ is also policy-adaptive submodular w.r.t. the prior distribution $p(\phi|\psi_S)$.*

*Proof.* Let $\psi_A$ and $\psi_B$ be partial realizations such that $\psi_A \subseteq \psi_B$ and $B \subseteq V \setminus S$. Let $\pi$ be any policy such that $\mathrm{range}(\pi) \subseteq V \setminus (S \cup B)$. From the definition, $\Delta_{f(\cdot|\psi_S)}(\cdot|\psi_A)$ is equal to $\Delta_f(\cdot|\psi_A \cup \psi_S)$. This leads to

$$\Delta_{f(\cdot|\psi_S)}(\pi|\psi_A) = \Delta_f(\pi|\psi_A \cup \psi_S) \geq \Delta_f(\pi|\psi_B \cup \psi_S) = \Delta_{f(\cdot|\psi_S)}(\pi|\psi_B).$$

The inequality is due to the policy-adaptive submodularity of $(f, p)$. It follows that $f(\cdot|\psi_S)$ is policy-adaptive submodular w.r.t. $p(\phi|\psi_S)$. $\qquad\square$

Define $E(\pi, \phi)$ to be the set of items obtained when $\pi$ is executed under realization $\phi$, that is, $E(\pi, \phi) = \mathrm{dom}(\psi(\pi, \phi))$. To prove the next lemma, we use the following lemma proved by Golovin and Krause [19].

**Lemma C.2** (See [19, Lamme 5.3]). *Suppose we have made observations $\psi$ after selecting $\mathrm{dom}(\psi)$. Let $\pi^*$ be any policy such that $|E(\pi^*, \phi)| \leq k$ for all $\phi$. Then for adaptive monotone submodular $f$*

$$\Delta(\pi^*|\psi) \leq \max_{A \subseteq E, |A| \leq k} \sum_{e \in A} \Delta(e|\psi).$$

Define $f_{\mathrm{avg}}(\pi) = \Delta(\pi|\emptyset)$ to be the average gain of a policy $\pi$. Here we prove the lemma about the maximal expected gain of a random selected subset without the policy-adaptive submodularity.

**Lemma C.3.** *Let $f : 2^{V \times \mathcal{Y}} \to \mathbb{R}_{\geq 0}$ be a set function and $p$ any prior over realizations such that $(f, p)$ is adaptive submodular and adaptive monotone. Let $\pi$ be any policy such that $|\psi(\pi, \phi)| \leq k$ for all $\phi$. Let $T \subseteq V$ be a random set that contains each item of $V$ independently with probability $r \in [0, 1]$. Then it holds that*

$$\mathbb{E}_T \left[ \max_{s \in T} \Delta(s|\emptyset) \right] \geq \frac{1 - e^{-rk}}{k} f_{\mathrm{avg}}(\pi).$$

*Proof.* From the adaptive data dependent bound (Lemma C.2), we have

$$f_{\text{avg}}(\pi) = \Delta(\pi|\emptyset) \le \max_{A \subseteq V, |A| \le k} \sum_{s \in A} \Delta(s|\emptyset).$$

Let $\{s_1, \cdots, s_k\} = \text{argmax}_{A \subseteq V, |A| \le k} \sum_{s \in A} \Delta(s|\emptyset)$. The probability that none of $s_1, \cdots, s_k$ is in $T$ is upper bounded as

$$\mathbb{P}_T[\forall j = 1, \cdots, k, \; s_j \notin T] = (1-r)^k \le \text{e}^{-rk}.$$

Under the condition that there is at least one $s_j$ in $T$, the expected value of $\max\{\Delta(s_j|\emptyset) \mid s_j \in T\}$ is no less than the average $\sum_{j=1}^k \Delta(s_j|\emptyset)/k$ because of the symmetry of $s_j$s. Since we have at least one $s_j$ with probability no less than $(1 - \text{e}^{-rk})$, it holds that

$$
\begin{aligned}
&\mathbb{E}_T[\max\{\Delta(s|\emptyset) \mid s \in T\}] \\
&= \mathbb{P}_T[\exists j = 1, \cdots, k, \; s_j \in T] \, \mathbb{E}_T[\max\{\Delta(s|\emptyset) \mid s \in T\} \mid \exists j = 1, \cdots, k, \; s_j \in T] \\
&\ge \frac{1 - \text{e}^{-rk}}{k} \sum_{j=1}^k \Delta(s_j|\emptyset) \\
&\ge \frac{1 - \text{e}^{-rk}}{k} f_{\text{avg}}(\pi).
\end{aligned}
$$

$\square$

Feige, Mirrokni, and Vondrák [15] gave the useful lemma about the value of a randomly selected subset for non-adaptive submodular functions as follows:

**Lemma C.4** (See [15, Lemma 2.2]). *Let $g : 2^X \to \mathbb{R}$ be submodular. Denote by $A(p)$ a random subset of $A$ where each element appears with probability $p$. Then*

$$\mathbb{E}[g(A(p))] \ge (1-p)g(\emptyset) + pg(A)$$

This lemma is a crucial part of many randomized algorithms for submodular maximization, but cannot be extended straightforward to adaptive submodular functions. However, assuming the policy-adaptive submodularity, we can prove a similar property for the adaptive setting. Before stating the lemma, we define a partially deleted policy tree.

**Definition C.5** (Grafted policy). *Let $\pi$ be an arbitrary policy, and let $T$ be a subset of $V$. Let $s$ denote the item that $\pi$ selects first, and $\pi_y$ the subpolicy executed by $\pi$ when $Y_s$ turns out to be $y$ for each $y \in \mathcal{Y}$. Formally, the grafted policy $\pi_T$ of $\pi$ for $T$ under partial realization $\psi$ is a randomized policy[6] defined recursively as follows: If $s \in T$, $\pi_T$ selects $s$ first, observes its label $y$ and proceeds to the grafted policy of $\pi_y$ for $T$ under partial realization $\psi \cup \{(s, y)\}$. If $s \notin T$, $\pi_T$ proceeds randomly with probability $\mathbb{P}[Y_s = y|\psi]$ to the grafted policy of $\pi_y$ for $T$ under partial realization $\psi \cup \{(s, y)\}$ for each $y \in \mathcal{Y}$. For simplicity, we omit to mention the partial realization if it is the grafted policy under no observation.*

Intuitively, the grafted policy is obtained by repeating the grafting operation, which is indicated in Figure 3, from the leaves to the root for each removed item.

Now we prove the adaptive version of Lemma C.4.

**Lemma C.6.** *Let $f : 2^{V \times \mathcal{Y}} \to \mathbb{R}$ be a set function and $p$ any prior over realizations such that $(f, p)$ is policy-adaptive submodular. Let $\pi$ denote an arbitrary policy. Let $T$ be a random subset of $V$ where each element appears with probability $r \in [0, 1]$. If $\pi_T$ is the grafted policy for $T$, it holds that*

$$\mathbb{E}_T[f_{\text{avg}}(\pi_T)] \ge (1-r)f(\emptyset) + rf_{\text{avg}}(\pi).$$

Figure 3: If item $s$ is not contained in restricted set $T$, remove $s$ from the original policy $\pi$ and proceed to the subpolicy, which is executed by $\pi$ when $Y_s$ turns out to be $y$, with probability $\mathbb{P}[Y_s = y|\psi]$. Repeating this operation recursively from the leaves to the root, we obtain the grafted policy of $\pi$.

*Proof.* We apply the induction on the height $k$ of the policy tree $\pi$ to prove the statement.

In the case of $k = 1$, the item $s \in T$ that $\pi$ selects is also selected by $\pi_T$ if $s \in T$. If $s \notin T$, no item is selected by $\pi_T$. Since the probability that $T$ contains $s$ is $r$, $\mathbb{E}_T[f_{\mathrm{avg}}(\pi_T)] = (1-r)f(\emptyset) + r f_{\mathrm{avg}}(\pi)$.

Assume the statement holds in the case of $k \leq l - 1$, we show it holds in the case of $k = l$. Let $\pi$ be any policy whose height is $l$. Let $s$ be the item $\pi$ selects first and $\pi_y$ the subpolicy executed by $\pi$ just after the label of $s$ turns out to be $y \in \mathcal{Y}$. Denoted by $\pi_{T,y}$ the grafted policy of $\pi_y$ for $T$ under the observation $\{(s, y)\}$.

Now we can decompose the expected total gain of the policy $\pi$ into the root and subtrees, i.e.,

$$f_{\mathrm{avg}}(\pi) = f(\emptyset) + \Delta(s|\emptyset) + \sum_{y \in \mathcal{Y}} p(\psi_y)\Delta(\pi_y|\psi_y) \tag{1}$$

where $\psi_y = \{(s, y)\}$ for each $y \in \mathcal{Y}$.

From Lemma C.1, $(f(\cdot|\psi_y), p(\cdot|\psi_y))$ is policy-adaptive submodular. The grafted policy $\pi_{T,y}$ under the observation $\psi_y$ is the same as the grafted policy of $\pi_y$ for $T$ under no observation with respect to $p(\phi|\psi_y)$. Since $f(\emptyset|\psi_y) = 0$, applying the induction hypothesis and we obtain that for all $y \in \mathcal{Y}$,

$$\mathbb{E}_T[\Delta(\pi_{T,y}|\psi_y)] \geq r\Delta(\pi_y|\psi_y). \tag{2}$$

We consider separately the cases where $s \in T$ and $s \notin T$. In the case of $s \in T$, it holds that

$$f_{\mathrm{avg}}(\pi_T) = f(\emptyset) + \Delta(s|\emptyset) + \sum_{y \in \mathcal{Y}} p(\psi_y)\Delta(\pi_{T,y}|\psi_y). \tag{3}$$

In the case of $s \notin T$, we have

$$f_{\mathrm{avg}}(\pi_T) = f(\emptyset) + \sum_{y \in \mathcal{Y}} p(\psi_y) f_{\mathrm{avg}}(\pi_{T,y})$$
$$\geq f(\emptyset) + \sum_{y \in \mathcal{Y}} p(\psi_y)\Delta(\pi_{T,y}|\psi_y). \tag{4}$$

The second inequality is due to the policy-adaptive submodularity of $(f, p)$.

Hence we obtain

$$
\mathbb{E}_T[f_{\mathrm{avg}}(\pi_T)] = \mathbb{P}[s \in T]\,\mathbb{E}_T[f_{\mathrm{avg}}(\pi_T) \mid s \in T] + \mathbb{P}[s \notin T]\,\mathbb{E}_T[f_{\mathrm{avg}}(\pi_T) \mid s \notin T]
$$

$$
\geq r\left(f(\emptyset) + \Delta(s|\emptyset) + \sum_{y \in \mathcal{Y}}\left(p(\psi_y)\mathbb{E}_T[\Delta(\pi_{T,y}|\psi_y) \mid s \in T]\right)\right)
$$

$$
+ (1-r)\left(f(\emptyset) + \sum_{y \in \mathcal{Y}}\left(p(\psi_y)\mathbb{E}_T[\Delta(\pi_{T,y}|\psi_y) \mid s \notin T]\right)\right) \qquad \text{(Since (3) and (4))}
$$

$$
= f(\emptyset) + r\Delta(s|\emptyset) + \sum_{y \in \mathcal{Y}} p(\psi_y)\mathbb{E}_T[\Delta(\pi_{T,y}|\psi_y)]
$$

$$
\geq f(\emptyset) + r\Delta(s|\emptyset) + \sum_{y \in \mathcal{Y}} p(\psi_y) r\Delta(\pi_y|\psi_y) \qquad \text{(Since (2))}
$$

$$
= (1-r)f(\emptyset) + r\left(f(\emptyset) + \Delta(s|\emptyset) + \sum_{y \in \mathcal{Y}} p(\psi_y)\Delta(\pi_y|\psi_y)\right)
$$

$$
= (1-r)f(\emptyset) + rf_{\mathrm{avg}}(\pi). \qquad \text{(Since (1))}
$$

$\square$

We illustrate adaptive submodularity is not enough to prove this lemma by providing an example that satisfies adaptive submodularity but not the inequality in Lemma C.6.

**Example C.7.** Let $f$ and $p$ be the same as Example A.6. Define a policy $\pi$ as follows. First it selects $a$ and if the state of $a$ is $-1$, it ends. If the state of $a$ is $+1$, it next selects $b$ and proceeds to select $c$ if the state of $b$ is $+1$. Consider the case of $r = 0.5$.

Let $T$ be a random subset of $\{a, b, c\}$. The case of $T = \{b, c\}$ is most important among eight cases. In this case the grafted policy $\pi_T$ selects $b$ with probability $0.1$, and if the state of $b$ is $+1$, it next selects $c$. The original policy $\pi$ selects $c$ with probability $0.09$ but the grafted policy $\pi_T$ selects $c$ with probability $0.01$. This difference results in $\mathbb{E}_T[f_{\mathrm{avg}}(\pi_T)] = 0.585$ and $(1-r)f(\emptyset) + rf_{\mathrm{avg}}(\pi) = 0.595$, which violates the inequality in Lemma C.6.

Lemma C.6 implies the following property about the optimal pool-based policy constructed in a randomly selected subset.

**Lemma C.8.** *Let $f : 2^{V \times \mathcal{Y}} \to \mathbb{R}_{\geq 0}$ be a set function and $p$ any prior over realizations such that $(f, p)$ is policy-adaptive submodular. Let $T$ be a random subset of $V$ where each element appears with probability $r \in [0, 1]$, and $\pi_V^*$ and $\pi_T^*$ the optimal pool-based policy selecting $k$ items from $V$ and $T$ respectively. Then it holds*

$$
\mathbb{E}_T[f_{\mathrm{avg}}(\pi_T^*)] \geq rf_{\mathrm{avg}}(\pi_V^*).
$$

*Proof.* Let $\pi_T$ be the grafted policy of $\pi_V^*$ for $T$. From Lemma C.6, we obtain

$$
\mathbb{E}_T[f_{\mathrm{avg}}(\pi_T)] \geq (1-r)f(\emptyset) + rf_{\mathrm{avg}}(\pi_V^*) \geq rf_{\mathrm{avg}}(\pi_V^*)
$$

The second inequality holds because of the non-negativity of $f$. Due to the optimality of $\pi_T^*$, it holds that $f_{\mathrm{avg}}(\pi_T^*) \geq f_{\mathrm{avg}}(\pi_T)$ for any $T \subseteq V$, which leads to the statement. $\square$

Suppose $\Phi \sim \psi$ means $\Phi$ follows the distribution $p(\Phi|\psi)$. Let $\pi' @ \pi$ denote the concatenated policy that run $\pi$ to completion and run $\pi'$ ignoring the information so far. To prove the main theorem we refer to the following lemma.

**Lemma C.9** (See [19, Lamme A.7]). *Fix a function $f : 2^E \times O^E \to \mathbb{R}_{\geq 0}$. Then $\Delta(e|\psi) \geq 0$ for all $\psi$ with $\mathbb{P}[\Phi \sim \psi] > 0$ and all $e \in E$ if and only if for all policies $\pi$ and $\pi'$, $f_{\mathrm{avg}}(\pi) \leq f_{\mathrm{avg}}(\pi' @ \pi)$.*

We prove the theorem about the theoretical guarantee for `AdaptiveStream` algorithm.

*Proof of Theorem 4.1.* Let $\sigma$ denote a random permutation in which the items arrive. Let $\pi_i^\sigma$ be the stream-based policy executed by running `AdaptiveStream` until it selects an item from the $i$th segment $S_i$ under the permutation $\sigma$ and then terminating. Let $\Psi_i^\sigma$ be a random partial realization obtained by $\pi_i^\sigma$.

First we bound from below the expected $i$th marginal gain $\mathbb{E}_\sigma[f_{\text{avg}}(\pi_i^\sigma) - f_{\text{avg}}(\pi_{i-1}^\sigma)]$ of the algorithm. Since at the $i$th step, the algorithm selects the item of the largest expected marginal gain in $S_i$,

$$\mathbb{E}_\sigma[f_{\text{avg}}(\pi_i^\sigma) - f_{\text{avg}}(\pi_{i-1}^\sigma)] = \mathbb{E}_{\sigma,\Phi}\left[\max_{s \in S_i} \Delta(s|\Psi_{i-1}^\sigma)\right]. \tag{5}$$

Fix $S_1 \cup S_2 \cup \cdots \cup S_{i-1}$ (formally, fix the part of a random permutation $\sigma$), we can regard $S_i$ as a random subset that contains each item $s \in V \setminus (S_1 \cup S_2 \cup \cdots \cup S_{i-1})$ independently with probability $1/(k-i+1)$. Let $T = V \setminus (S_1 \cup S_2 \cup \cdots \cup S_{i-1})$. Since $f(\cdot|\Psi_{i-1}^\sigma)$ is adaptive submodular w.r.t. $p(\phi|\Psi_{i-1}^\sigma)$, from Lemma C.3 we obtain

$$\mathbb{E}_{\sigma,\Phi}\left[\max_{s \in S_i} \Delta(s|\Psi_{i-1}^\sigma)\right] \geq \frac{1 - \exp(-\frac{k}{k-i+1})}{k} \mathbb{E}_{\sigma,\Phi}\left[\Delta(\pi_T^*|\Psi_{i-1}^\sigma)\right] \tag{6}$$

where $\pi_T^*$ is the optimal pool-based policy selecting $k$ items from $T$.

Under any fixed $\sigma$, we can view $\mathbb{E}_\Phi[\Delta(\pi_T^*|\Psi_{i-1}^\sigma)]$ as the expected gain of $\pi_T^*$ after $\pi_i^\sigma$ is executed, then it holds that

$$\begin{aligned}\mathbb{E}_\Phi\left[\Delta(\pi_T^*|\Psi_{i-1}^\sigma)\right] &= f_{\text{avg}}(\pi_T^* @ \pi_{i-1}^\sigma) - f_{\text{avg}}(\pi_{i-1}^\sigma) \\ &\geq f_{\text{avg}}(\pi_T^*) - f_{\text{avg}}(\pi_{i-1}^\sigma).\end{aligned} \tag{7}$$

The inequality follows from the adaptive monotonicity of $(f, p)$ and Lemma C.9.

Let $\pi_V^*$ be the optimal pool-based policy selecting $k$ items from $V$. Since $T$ can be regarded as a random subset that contains each item of $V$ independently with probability $(k - i + 1)/k$, from Lemma C.8 we have

$$\mathbb{E}_\sigma[f_{\text{avg}}(\pi_T^*)] \geq \frac{k - i + 1}{k} f_{\text{avg}}(\pi_V^*). \tag{8}$$

From (5), (6), (7), and (8), we obtain

$$\begin{aligned}\mathbb{E}_\sigma[f_{\text{avg}}(\pi_i^\sigma) - f_{\text{avg}}(\pi_{i-1}^\sigma)] &\geq \frac{1 - \exp(-\frac{k}{k-i+1})}{k}\left\{\frac{k - i + 1}{k} f_{\text{avg}}(\pi_V^*) - \mathbb{E}_\sigma[f_{\text{avg}}(\pi_{i-1}^\sigma)]\right\} \\ &\geq \frac{(1 - e^{-1})(k - i + 1)}{k^2} f_{\text{avg}}(\pi_V^*) - \frac{1}{k}\mathbb{E}_\sigma[f_{\text{avg}}(\pi_{i-1}^\sigma)].\end{aligned}$$

The upper and lower bounds $0 \leq \exp(-\frac{k}{k-i+1}) \leq e^{-1}$ are used in the second inequality.

We use the proof by contradiction to give the performance guarantee of `AdaptiveStream`. Assume $\mathbb{E}_\sigma[f_{\text{avg}}(\pi_k^\sigma)] < \alpha f_{\text{avg}}(\pi_V^*)$ (the concrete value of $\alpha$ is specified later). Due to the adaptive monotonicity of $(f, p)$, we know that $\mathbb{E}_\sigma[f_{\text{avg}}(\pi_i^\sigma)]$ is increasing monotonically as $i$ increases. Hence for all $i = 1, \cdots, k$, it holds that $\mathbb{E}_{\sigma,\Phi}[f_{\text{avg}}(\pi_i^\sigma)] < \alpha f_{\text{avg}}(\pi_V^*)$.

Let $l = \lfloor(\sqrt{3} - 1)k\rfloor$. Taking the summation of the above inequality until the $l$th step, we obtain the following:

$$\begin{aligned}\mathbb{E}_\sigma[f_{\text{avg}}(\pi_l^\sigma)] &\geq \frac{1 - e^{-1}}{k^2} \sum_{i=1}^l (k - i + 1) f_{\text{avg}}(\pi_V^*) - \frac{1}{k} \sum_{i=1}^l \mathbb{E}_\sigma[f_{\text{avg}}(\pi_{i-1}^\sigma)] \\ &> \frac{1 - e^{-1}}{k^2} \sum_{i=1}^l (k - i + 1) f_{\text{avg}}(\pi_V^*) - \frac{1}{k} \sum_{i=1}^l \alpha f_{\text{avg}}(\pi_V^*).\end{aligned}$$

Similarly, taking the summation until the $(l + 1)$th step, we obtain

$$\mathbb{E}_\sigma[f_{\text{avg}}(\pi_{l+1}^\sigma)] > \frac{1 - e^{-1}}{k^2} \sum_{i=1}^{l+1} (k - i + 1) f_{\text{avg}}(\pi_V^*) - \frac{1}{k} \sum_{i=1}^{l+1} \alpha f_{\text{avg}}(\pi_V^*).$$

Since $\mathbb{E}_\sigma[f_{\mathrm{avg}}(\pi_l^\sigma)]$ and $\mathbb{E}_\sigma[f_{\mathrm{avg}}(\pi_{l+1}^\sigma)]$ are at most $\alpha f_{\mathrm{avg}}(\pi_V^*)$ from the assumption, we have

$$2\alpha f_{\mathrm{avg}}(\pi_V^*) > \frac{1 - \mathrm{e}^{-1}}{k^2}\left\{2\sum_{i=1}^{l}(k - i + 1) + (k - l)\right\}f_{\mathrm{avg}}(\pi_V^*) - 2(\sqrt{3} - 1)\alpha f_{\mathrm{avg}}(\pi_V^*)$$

$$\geq \left\{(4\sqrt{3} - 6)(1 - \mathrm{e}^{-1}) - 2(\sqrt{3} - 1)\alpha\right\}f_{\mathrm{avg}}(\pi_V^*).$$

If $\alpha \leq (2 - \sqrt{3})(1 - \mathrm{e}^{-1})$, it leads to the contradiction. Therefore we conclude that

$$\mathbb{E}_\sigma[f_{\mathrm{avg}}(\pi_k^\sigma)] \geq (2 - \sqrt{3})(1 - \mathrm{e}^{-1})f_{\mathrm{avg}}(\pi_V^*) \geq 0.16 f_{\mathrm{avg}}(\pi_V^*).$$

□

Next we prove the theorem about the theoretical guarantee for `AdaptiveSecretary` algorithm.

*Proof of Theorem 4.2.* Similarly as the stream setting, let $\pi_i^\sigma$ be the stream-based policy obtained by running `AdaptiveSecretary` until it selects an item from or passes the $i$th segment $S_i$ under the permutation $\sigma$ and then terminating. Let $\Psi_i^\sigma$ be a random partial realization obtained by $\pi_i^\sigma$. The outline is completely the same as the proof of Theorem 4.1, but the lower bound of the expected marginal gain at each step is different.

The adaptive secretary algorithm uses the classical secretary algorithm as a subroutine. It selects the item of the maximal expected marginal gain at least with probability $1/\mathrm{e}$ from each segment, i.e.,

$$\mathbb{E}_\sigma[f_{\mathrm{avg}}(\pi_i^\sigma) - f_{\mathrm{avg}}(\pi_{i-1}^\sigma)] \geq \mathbb{E}_{\sigma,\Phi}\left[\frac{1}{\mathrm{e}}\max_{s\in S_i}\Delta(s|\Psi_{i-1}^\sigma)\right].$$

The lower bound of the expected marginal gain at each step is

$$\mathbb{E}_\sigma[f_{\mathrm{avg}}(\pi_i^\sigma) - f_{\mathrm{avg}}(\pi_{i-1}^\sigma)] \geq \frac{(1 - \mathrm{e}^{-1})(k - i + 1)}{\mathrm{e}k^2}f_{\mathrm{avg}}(\pi_V^*) - \frac{1}{\mathrm{e}k}\mathbb{E}_\sigma[f_{\mathrm{avg}}(\pi_{i-1}^\sigma)].$$

Applying the similar evaluation to this inequality with $l = \lfloor(-\mathrm{e} + \sqrt{\mathrm{e}^2 + 2\mathrm{e}})k\rfloor$, we conclude that

$$\mathbb{E}_\sigma[f_{\mathrm{avg}}(\pi_k^\sigma)] \geq \frac{1 - \mathrm{e}^{-1}}{2\mathrm{e}\sqrt{1 + 2/\mathrm{e}}}f_{\mathrm{avg}}(\pi_V^*) \geq 0.08 f_{\mathrm{avg}}(\pi_V^*).$$

□