[Reviews · NeurIPS 2016]

Reviewer 1

Summary

Motivated by active learning, the paper presents algorithms for adaptive submodular maximization subject to a cardinality constraint, in the streaming and secretary settings. (Both algorithms process a stream of examples in random order and are limited to query at most k labels. The streaming setting allows the algorithm to keep a bounded-size buffer of examples that have been observed but not yet labeled. The secretary setting disallows such deferred decisions.) The algorithms are very simple. Both of them generalize the algorithm of Bateni et al. for the submodular secretary problem in the obvious way: divide the input stream into k contiguous blocks of size n/k, and simply try to pick the element of each block that makes the biggest marginal contribution, given previously labeled elements. The performance guarantees for these algorithms are fairly small constant-factor approximation guarantees (0.15 and 0.08, respectively) proven under a novel assumption called "policy-adaptive submodularity" which generalizes adaptive submodularity in the natural way to this context, and which seems like a plausible assumption to make for the intended applications.

Qualitative Assessment

In my opinion, the marginal contribution of the paper beyond prior work on adaptive submodularity and the submodular secretary problem is too weak to warrant inclusion in NIPS. The streaming setting is well-motivated for this type of problem, but paper's technical contribution strikes me as too incremental. A question about the experiments (Section 5): what set function f was used as the submodular function in the experiments? The paper doesn't seem to explain this point.

Confidence in this Review

2-Confident (read it all; understood it all reasonably well)


Reviewer 2

Summary

The paper proposes the class of policy-adaptive submodular functions, a subclass of adaptive submodular functions, and shows that this new class of functions includes some existing utilities for pool-based active learning. The paper then explores the use of this class in stream-based active learning and proposes two algorithms for the stream and secretary settings. Both algorithms are shown to have theoretical performance guarantees compared to the optimal pool-based algorithm.

Qualitative Assessment

Although the notion of policy-adaptive submodular functions is interesting, I think it is only a marginal improvement from adaptive submodularity. Theorems 4.1 and 4.2 in the paper make sense, although the constant factors (0.16 and 0.08) are quite small. Does the small constants come from the fact that we compare the stream-based algorithms with the optimal pool-based algorithm? I think it would be more useful to also compare the proposed algorithms with the optimal stream-based algorithm. For example, I feel that we may be able to compare the AdaptiveStream algorithm with the optimal stream-based algorithm that selects one item from each of the k segments, even with only adaptive submodular utility functions. I think the paper should spend more space to explain their new definition, theorems and proof ideas. The pseudo-codes in Algorithms 1 and 2 can be simplified to make more space. I think the current manuscript lacks some insightful discussions of the new definition and theorems. For the experiments, I think it would also be useful to compare the proposed algorithms with other stream-based active learning algorithms, not just those restricted to the stream or secretary settings in this paper.

Confidence in this Review

2-Confident (read it all; understood it all reasonably well)


Reviewer 3

Summary

The paper proposes a new class of utility functions: policy-adaptive submodular functions. It provides a general framework based on policy-adaptive submodularity that can adapt existing pool-based methods to stream-based settings. The authors also give theoretical guarantees on their performance.

Qualitative Assessment

This is an interesting paper. The problem studied in the paper can be influential to active learning field. The paper is well-presented and easy to follow. However, the technical part is rather incremental. Some comments are listed below that need to be clarified. 1. The figure 3 is too hard to read. 2. What are the differences between the proposed framework with existing methods of submodular maximization in the stream and secretary settings? How their performance would compare with the proposed framework?

Confidence in this Review

2-Confident (read it all; understood it all reasonably well)


Reviewer 4

Summary

This paper studies the budgeted active learning problem (as under a cardinality constraint) in the stream-based setting. The paper assumes that data arrives in a stream in a RANDOM order, and provide analysis for both the steam and secretary settings, when the objective function satisfies a notion called policy-adaptive submodularity. This property is an extension of submodularity to the adaptive setting, but stricter than adaptive submodularity (in the sense that it considers the gain of a policy as opposed to the gain of a test; however in the adaptive setting a partial realization changes not only the distribution of a test outcome, but also the structure of a policy, and this policy-adaptive submodularity imposes a stronger constraint). The authors show that several adaptive submodular objective functions in active learning also satisfies policy-adaptive submodualriy, and then proposes algorithms stream-based learning. Finally the results are backed-up by the evaluation of the stream and secretary algorithms against pool-based algorithm on UCI WDBC and MNIST dataset.

Qualitative Assessment

Pros: The presentation is mostly clear. The paper shows that one can apply the proposed streaming algorithm, without changing the commonly used objective functions used in pool-based active learning setting. Proofs are sound, and experimental results show that the proposed algorithms work reasonably well in comparison with the pool-based setting. Cons: The stream-based adaptive sensor placement application does not appear convincing to me. Is the condition $range(\pi) \subseteq V \setminus B$ in Def 3.1 necessary? Policy-adaptive submodularity is used for providng a lower bound on the expected gain of a policy on a random sequence of data points (Lemma B.7). It would be useful and more insightful if the authors can provide intuition in the main paper of why adaptive submodularity is not sufficient for the proof. Is policy-adaptive submodilarity (provably) necessary? or are there examples of adaptive submodular functions (which are not policy-adaptive submodular) that can not achieve the provided theoretical guarantee? Data arriving at a random order seems to be a strict constraint. It's a necessary assumption for the (adaptive secretary) algorithm to work provably well, because the algorithm discards the first 1/e data points in each segment, which can contain the most beneficial points in each segment. The authors point out the need of a new algorithm when the data arrives in a fixed order; it would be helpful (for completeness) to provide some very brief discussion on this (i.e., what applications fall into this category; why it's challenging; if there is any insight, or perhaps to point possible directions for solving this problem). Editorial issues: Example A.6, There are two joint probabilities of P[Ya = +1, Yb = +1]. One of them should be P[Ya = +1, Yb = -1]. Ln 103, k labeled instances of the high utility --> k labeled instances of high utility Lemma B.4, Let f: .. --> Let g: ...

Confidence in this Review

3-Expert (read the paper in detail, know the area, quite certain of my opinion)


Reviewer 5

Summary

This paper studies the active learning problem under the streaming setting. The authors propose a class of policy-adaptive submodular functions, which is shown to be satisfied by some existing utility functions people define for active learning. Two algorithms are given for optimizing the policy-adaptive submodular functions under the streaming and secretary setting. However, the proposed algorithms as well as their analysis are very similar to Bateni et al. 2013.

Qualitative Assessment

I think the paper is overall well-written and the streaming active learning problem considered here is interesting. My biggest concern is the novelty of this work: the proposed algorithm AdaptiveSecretary is almost identical to the submodular secretary algorithm proposed in Bateni et al. 2013, so is the analysis. AdaptiveStream, on the other hand, is only trivial modification of AdaptiveSecretary, so is its analysis. My current assessment of the paper is borderline. I am more willing to accept the paper if more empirical results are available. For example, (1) the current paper considers only one submodular utility function (AluMA), and it would be interesting to show comparison of the proposed algorithms in terms of many other adaptive submodular utility functions. (2) baselines other than uncertainty sampling are needed. (3) it would also be interesting to test classifiers other than SVM. One minor suggestion: Line 92-99 may be removed since it is a bit repetitive with previous sections.

Confidence in this Review

2-Confident (read it all; understood it all reasonably well)


Reviewer 6

Summary

The paper proposes a general framework that can convert any pool-based active learning method to a stream-based active learning method when the objective function that represents the informativeness of sets of data is monotone and policy-adaptive submodular. Policy-adaptive submodular functions, which are an extension of adaptive submodular functions, are defined in the paper as set functions that their expected marginal gain of a subset is more than expected marginal gain of its superset according to a policy of adding items. Two simple algorithms for stream setting and secretary setting are proposed. These algorithms guaranty constant approximation factors of the optimal solution. The presented algorithms are compared with uncertainty sampling methods on two datasets.

Qualitative Assessment

The paper is well written and is innovative on the definition of policy-adaptive submodularity. Despite the interesting results of the theoretical part, the experimental section is weak. Only one pool-based method and its corresponding stream-based algorithms are considered and the comparisons are just on two datasets.

Confidence in this Review

2-Confident (read it all; understood it all reasonably well)